# Biomechanical Loading Comparison between Titanium and Unsintered Hydroxyapatite/Poly-L-Lactide Plate System for Fixation of Mandibular Subcondylar Fractures

**DOI:** 10.3390/ma12091557

**Published:** 2019-05-13

**Authors:** Shintaro Sukegawa, Takahiro Kanno, Norio Yamamoto, Keisuke Nakano, Kiyofumi Takabatake, Hotaka Kawai, Hitoshi Nagatsuka, Yoshihiko Furuki

**Affiliations:** 1Department of Oral and Maxillofacial Surgery, Kagawa Prefectural Central Hospital, 1-2-1, Asahi-machi, Takamatsu, Kagawa 760-8557, Japan; furukiy@ma.pikara.ne.jp; 2Department of Oral and Maxillofacial Surgery, Shimane University Faculty of Medicine, Shimane 693-8501, Japan; tkanno@med.shimane-u.ac.jp; 3Department of Orthopaedic Surgery, Kagawa Prefectural Central Hospital, Takamatsu, Kagawa 761-0396, Japan; lovescaffe@yahoo.co.jp; 4Department of Oral Pathology and Medicine, Okayama University Graduate School of Medicine, Dentistry and Pharmaceutical Sciences, Okayama 7008530, Japan; pir19btp@okayama-u.ac.jp (K.N.); gmd422094@s.okayama-u.ac.jp (K.T.); de18018@s.okayama-u.ac.jp (H.K.); jin@okayama-u.ac.jp (H.N.)

**Keywords:** mandibular condylar fracture, unsintered hydroxyapatite/poly-l-lactide composite plate, bioactive resorbable plate, biomechanical loading evaluation, fracture fixation

## Abstract

Osteosynthesis absorbable materials made of uncalcined and unsintered hydroxyapatite (u-HA) particles, poly-l-lactide (PLLA), and u-HA/PLLA are bioresorbable, and these plate systems have feasible bioactive osteoconductive capacities. However, their strength and stability for fixation in mandibular subcondylar fractures remain unclear. This in vitro study aimed to assess the biomechanical strength of u-HA/PLLA bioresorbable plate systems after internal fixation of mandibular subcondylar fractures. Tensile and shear strength were measured for each u-HA/PLLA and titanium plate system. To evaluate biomechanical behavior, 20 hemimandible replicas were divided into 10 groups, each comprising a titanium plate and a bioresorbable plate. A linear load was applied anteroposteriorly and lateromedially to each group to simulate the muscular forces in mandibular condylar fractures. All samples were analyzed for each displacement load and the displacement obtained by the maximum load. Tensile and shear strength of the u-HA/PLLA plate were each approximately 45% of those of the titanium plates. Mechanical resistance was worst in the u-HA/PLLA plate initially loaded anteroposteriorly. Titanium plates showed the best mechanical resistance during lateromedial loading. Notably, both plates showed similar resistance when a lateromedially load was applied. In the biomechanical evaluation of mandibular condylar fracture treatment, the u-HA/PLLA plates had sufficiently high resistance in the two-plate fixation method.

## 1. Introduction

Mandibular condylar fractures constitute 25%–35% of all mandibular fractures [1] and are the most common form of mandible fracture.

The selection between closed (conservative treatment) and open (surgical treatment) techniques are linked to the type of fracture, the patient’s age, and the degree of functional impairment caused by the fracture. Growing patients with certain undisplaced mandibular condylar fractures can undergo closed treatment, but severely displaced fractures are indications for open surgical reduction and fixation with internal rigid devices, generally with plates and screws [2]. In adult patients, the condylar head can be adjusted for functional adaptation of the condyle without restoration of the anatomy caused by displaced base fractures with a loss of ramus height [1]. Therefore, high condylar fractures with little bone available for fixation are mostly treated non-surgically. On the other hand, lower condylar fractures that affect the subcondyle, condylar base, and condylar neck are generally treated by open reduction and internal fixation [3,4]. The surgical treatment of mandibular condylar fractures involves fixation of the fractured stumps with the use of plates and screws. There are several types of fixation for fractures of the mandibular condyle [5]; the most common fixation technique is to use two four-hole straight plates [6]. In general, a titanium metal plate system has been used as a standard osteosynthesis material.

However, titanium metal plates have been reported to produce long-term problems related to implantation in the human body, such as intracranial migration, growth retardation of the craniofacial skeleton, and hypersensitivity to cold. In recent years, as an alternative to metal plates, resorbable plates, which do not require plate removal, have been used as osteosynthesis material in maxillofacial surgery [7]. Bioresorbable osteosynthesis devices have been evolving, with improvements in bioresorbability and marked bioactivity with new material compositions, for different and better in situ behavior. Currently, many osteosynthesis absorbable materials are commercially available; these include OSTEOTRANS-MX (known in Japan as Super FIXSORB MX^®^; Teijin Medical Technologies Co., Ltd. Osaka, Japan). These bioactive and bioresorbable devices are made from composites of uncalcined and unsintered hydroxyapatite (u-HA) particles and poly-l-lactide (PLLA). This u-HA/PLLA osteosynthesis material is an excellent bioactive device that has an osteoconductive ability in the short-term and absorbs safely and reliably in the long-term [8,9]. However, both the strength and stability of this plate system for fixation to a mandibular subcondylar fracture are still unclear. Thus, the aim of this in vitro study was to assess the biomechanical strength of u-HA/PLLA bioactive resorbable plate systems after internal fixation of mandibular subcondylar fractures.

## 2. Materials and Methods

### 2.1. Materials

We used the bioresorbable Super FIXSORB MX^®^ (Teijin Medical Technologies Co., Ltd., Osaka, Japan) osteosynthesis system. Forged composites of u-HA/PLLA were processed by machining or milling treatments to form various miniscrews and miniplates, which contained 30- and 40-weight fractions of u-HA particles (raw HA, neither calcined nor sintered material), respectively, in composites (hereinafter referred to as u-HA 30 miniscrews and u-HA 40 miniplates).

A 2.0 mm miniplate system (MatrixMANDIBLE™ Adaption Plate; DePuy Synthes, Raynham, Mass.) (Johnson & Johnson, New Brunswick, USA) was used as the titanium metal plate.

### 2.2. Evaluation of Tensile and Shear Strength of Titanium and Bioresorbable Plate Systems

#### 2.2.1. Sample Preparation

We prepared mechanical strength models by affixing the titanium and u-HA/PLLA plates to polyetherketoneketone (PEKK) plates (thickness 3 mm) with screws. The PEKK plates were held in place by different osteosynthesis systems. Plates and screws were used to form the following groups:A single u-HA/PLLA bioresorbable straight plate, with the plate held in place on each side with two screws (total four screws);A single titanium straight plate, with the plate held in place on each side with two screws (total four screws).

In this study, the bioresorbable osteosynthesis material consisted of 1.4 mm-thick plates and screws (diameter 2 mm; length 6 mm). In comparison, the titanium osteosynthesis material consisted of 1.0 mm-thick plates and screws (diameter 2 mm; length 6 mm). The experimental fixed models were mounted on an Autograph AG-20kNXD test frame (Shimadzu Co., Kyoto, Japan) across the chuck. The maximum stress until the plate or screws were destroyed and the stress at the time of 1 mm movement were measured, and the load was applied at a test speed of 10 mm/min. Two types of strength tests (the tensile and shear strength tests) were performed five times each.

#### 2.2.2. Strength Measurement

Using the method illustrated in Figure 1A according to the Japanese Industrial Standard K7113, we measured tensile strength. The peak value of the load profile attained by the Autograph AG-20kNXD (Shimadzu Co., Kyoto, Japan) was considered the tensile strength. Using the method illustrated in Figure 1B according to the Japanese Industrial Standard K7113, we measured shear strength. The peak value of the load profile attained by the Autograph AG-20kNXD was considered the shear strength.

### 2.3. Biomechanical Loading Evaluation

#### 2.3.1. Sample Preparation

We used 20 polyurethane replicas of human hemimandibles (Mandible, Code #8900, SYNBONE AG, Laudquart, Tardisstrasse, Switzerland). Although polyurethane mandible replicates the property of spongy bone [10], this model is useful to obtain preliminary results concerning the stability of the investigated osteosynthesis systems. An ordinary subcondylar fracture model was investigated by using the osteotomy line connecting the mandibular notch to the midpoint of the mandibular ramus’ posterior border. We created a fracture in the right condyle of each model, mimicking the experimental method of Meyer et al. [11]. Initially, these fracture models were created with the use of a cutting guide by a computer-controlled program. A partial cut was made in each hemimandibular model with a diamond disk (KG Sorensen, Cotia, São Paulo, Brazil). A silicone mold of the subcondylar area of the osteosynthesis was made on the mandible and drilled to form guide holes, and then, using the mold as a guide, we completed the cut. This method was used to create identical cuts and perforations in all the hemimandibular models, thus guiding the positions for plate fixation according to each study group.

#### 2.3.2. Fixation Method for Mandibular Condylar Fracture

We used two straight miniplates along the ideal line of the mandibular condyle as proposed by Meyer et al. [12]. The technique is to plate below the mandibular notch and below the mandibular posterior border, respectively. This technique with monocortical screws had proved to be the most reliable and functionally stable osteosynthesis procedure for subcondylar fractures. The bone segments were held in place by different commercially available osteosynthesis methods (a titanium miniplate and bioresorbable plate) and monocortical screws. With both types of plates, bone fixation could be performed without plate bending. Thus, we created two conditions (Figure 2A): 

(1) In each bone segment in group A, double titanium straight plates (MatrixMANDIBLE™ Adaption Plate, 1.0 mm thick) with four monocortical screws (2.0 mm in diameter and 6 mm long) were installed; and (2) in each bone segment in group B, double-u-HA/PLLA straight plates (Super FIXSORB MX ^®^, 1.4 mm thick), each with four monocortical screws (2.0 mm in diameter and 6 mm long), were installed.

We used a total of 20 hemimandibular replicas, and 10 were allocated to each of the two groups.

#### 2.3.3. Biomechanical Loading Test

After bone segment fixation, the replicas were mounted on a testing machine (AG-20KNX; Shimazu) (Shimadzu Co., Kyoto, Japan), which was based on a biomechanical cantilever-bending model that simulates masticatory forces. The mandibular body and angle areas of each replica were then stabilized. Adaptation of the polyurethane hemimandible to the machine was guaranteed through a metallic support. In reference to past biomechanical evaluation methods, the replicas were then subjected to linear loading in two directions: from anterior to posterior (vertically; Figure 2B) and from lateral to medial (horizontally; Figure 2C). These forces simulated the muscular forces applied to an actual fractured condyle. The material testing unit created linear displacement at a rate of 1 mm/min, and loading continued until the maximum load was reached. The peak load and displacement for each replica were recorded. All replicas were analyzed for 0.5, 1.0, 1.5, 2.0, 3.0, and 5.0 mm displacements by loading and for the amount of displacement by the maximum load. Means and standard deviations were derived and assessed for statistical significance.

### 2.4. Statistical Analysis

Data were recorded and entered into an electronic database during the course of the evaluation by means of Microsoft Excel (Microsoft, Inc., Redmond, USA). Means and standard deviations were used for normal data distributions. The database was transferred to JMP version 11.2 for Macintosh computers (SAS Institute, Inc., Cary, USA) for statistical analysis. Then, t-tests for independent samples were performed to investigate whether there were significant differences between the mean values of the groups. The level of significance was set at *p* < 0.05.

## 3. Results

### 3.1. Tensile and Shear Strength Evaluation

The results are shown in Figure 3. In the tensile test, the mean maximum test forces were 806.1 N (standard deviation 7.9 N) in group A (titanium plates fixed with titanium screws) and 208.8 N (standard deviation 10.3 N) in group B (u-HA/PLLA bioresorbable plates with bioresorbable screws). The mean test forces at 1 mm displacement were 382.1 N (standard deviation 19.5 N) in group A and 181.8 N (standard deviation 4.2 N) in group B. The mean maximum tensile test force in group B was 25.9% that of group A, and the test force at 1 mm displacement in group B was 47.6% that of group A. These differences were significant (*p* < 0.05) (Figure 3A).

In the shear test, the mean maximum test force and the test force at 1 mm displacement were 390.5 N (standard deviation 11.0 N) and 130.6 N (standard deviation 12.2 N), respectively, in group A. In group B, these mean forces were 93.2 N (standard deviation 4.6 N) and 58.1.8 N (standard deviation 12.2 N), respectively. The maximum shear test force in group B was 23.9% that of group A, and the test force at 1 mm displacement in group B was 44.5% that of group A.

The strength of the titanium-based fixation system was obviously higher than that of the u-HA/PLLA bioresorbable system in both the tensile test and the shear test. In the shear test, the titanium plates were significantly stronger (*p* < 0.05) than the u-HA/PLLA bioresorbable plates at maximum shear test force and at 1 mm displacement test force (Figure 3B).

### 3.2. Biomechanical Loading Evaluation

The results of this experiment show that the mechanical resistances among the osteosynthesis devices remained proportional to the amount of displacement. In the vertical loading test, the titanium plates had a significantly higher load value than did the u-HA/PLLA bioresorbable plates at displacements of 0.5 and 1 mm. In the displacements of 1.5–5 mm, there was no significant difference between the two types of plates (Figure 4). As the load in the anteroposterior direction increased, although the bone fragments fixed by the anterior plate were separated, the position of the posterior bone fragments did not change (Figure 5).

In contrast, when the forces were applied to the condyle in the lateromedial direction, there was no significant difference between the u-HA/PLLA bioresorbable plates and the titanium plates at all displacements (Figure 6). As the load in the lateromedial direction increased, there was no excessive load on either titanium or u-HA/PLLA bioresorbable anterior and posterior plates, and both types of plates in both locations were flexed (Figure 7).

## 4. Discussion

Bioresorbable materials have been used clinically for osteosynthesis in various fields of maxillofacial surgery such as orthodontic surgery [13], repair of craniomaxillofacial fractures [7], bone augmentation for dental implantation [14], and reconstruction in situations of maxillofacial cysts and tumors. In traumatology, bioresorbable plates are used mainly in the treatment of orbital and zygomatic fractures [15,16,17] and in the treatment of mandibular fractures [8,18]. However, there are very few clinical studies of the treatment of mandibular subcondylar fractures with resorbable plates, and the number of cases reported is often low [19]. Bioresorbable material, unlike titanium plates, generally cannot resist high mechanical loads. Therefore, for mandibular condylar fracture treatment, it is necessary to know how much load the u-HA/PLLA bioresorbable plates can withstand in comparison with titanium plates. We aimed to clarify the strength of the bioresorbable plate system and the titanium plate system and to examine the possibility of the treatment choice of the osteosynthesis material for mandibular subcondylar fracture.

First, the mechanical strengths of the titanium and u-HA/PLLA bioresorbable plate systems were examined. The strength of the u-HA/PLLA bioresorbable system, in either tension or shear, was only approximately 45% that of the titanium system. The u-HA/PLLA bioresorbable plate strength was improved slightly when the form of the plates was modified. This is comparable to previous research on the strength of titanium plates when the same material of the bioresorbable plate was used. However, our results were similar to those of previous research [20]. These were very important findings. For pure plate strength, these results suggest that the u-HA/PLLA bioresorbable plate system was much weaker than the titanium plate system.

However, the results of our biomechanical loading tests were very interesting. In the loading tests, we reproduced mandibular subcondylar fracture treatment with two straight miniplates along the ideal line of the mandibular condyle. In the lateromedial loading test, there was no significant difference between the u-HA/PLLA bioresorbable plates and the titanium plates. One of the most important clinical indications for surgical treatment of mandibular condylar fracture is the medial displacement of the condyle [21], which is caused by traction of the lateral pterygoid muscle. The primary function of the lateral pterygoid muscle is to pull the head of the condyle out of the mandibular fossa along the articular eminence to make the mandible protrude [22]. The lateral pterygoid muscle allows lateral movement of the mandible. This test simulates the lateral movements of mastication, and the strength of the bioresorbable plate system was sufficient. In contrast, in the anteroposterior loading test, the load value at the initial displacements of 0.5 and 1.0 mm was significantly larger in the titanium plates than in the u-HA/PLLA bioresorbable plates; the load in the latter was approximately 80% that of the titanium plates. The u-HA/PLLA bioresorbable plates were much weaker than the titanium plates. However, proper plate placement for mandibular subcondylar fracture treatment greatly improved the strength of u-HA/PLLA bioresorbable plates.

Champy and Lodde [23] advocated “functionally stable osteosynthesis” or “dynamic osteosynthesis,” according to which the plates must be placed along physiological tension lines that appear during the fracture. This concept was based on the theory that stabilization depends mainly on the compression of the fracture site until function is restored. Meyer et al. [11] demonstrated the presence of the compression line on the mandibular posterior border and the tension line on the mandibular notch in the mandibular condylar process. One posterior plate is located on the compression strain lines when placed along the condylar neck axis. This plate enables functionally stable osteosynthesis. The other plate, which enables dynamic osteosynthesis, should be on the traction line. To conform to dynamic osteosynthesis principles, the plate should be placed higher and more obliquely, parallel to the mandibular notch [11]. In short, the plate located at the traction line resists the dynamic force of the occlusion. The key to anteroposterior direction load testing is the presence of plates on the traction line.

In our experiment, as the anteroposterior load increased, the bone opening on the traction line increased. This opening was influenced by the strength of the plate on the traction line and was thought to cause the difference in load values of the titanium plates. Load values at displacements of 1.5 mm or more in the bioresorbable plates did not differ significantly from those in the titanium plate group. Therefore, we believe that the bioresorbable plate can serve clinically in osteosynthesis of mandibular subcondylar fractures. However, the mechanical model used for this study was a polyurethane model, which is softer than the actual mandible [9]. Hence, considering a slight decrease in the biomechanical strength of the bioresorbable model compared with the titanium equivalent, in clinical use of the bioresorbable plate system, early overloading soon after surgery should be avoided. A young patient with strong occlusal force also needs dietary instruction, such as eating soft foods after surgery.

At present, resorbable plate systems are used widely, and various resorbable plate systems have been developed on the basis of PLLA. Like the u-HA/PLLA plate system used in this study, these plate systems composed of various materials, including pure PLLA [15] or polyglycolic acid/PLLA [24], are commercially available. In particular, the u-HA/PLLA bioresorbable plate system that we studied has a very large biological advantage. Resorbable plates composed of pure PLLA are very strong [25], however, the strength of plates is reduced when PLLA is mixed with other materials. Unfortunately, PLLA osteosynthetic devices have several disadvantages, including lower dynamic strength, an inability to bond directly to bones, unstable resorption and decomposition processes, and long replacement times [26,27,28]. In contrast, u-HA/PLLA osteosynthesis devices significantly improved biological activity. It has been reported that u-HA/PLLA material induced no apparent inflammatory or foreign body reactions after implantation, and bonded directly to the human bone quickly [14]. Clinical studies have also clearly shown the effective osteoconductivity of the u-HA/PLLA plate system in maxillofacial treatment [8,29]. In addition, the reduction in strength due to the use of composite material was overcome by the unique compression molding and machining treatment in this osteosynthesis device [30]. The early bioactivity, such as osteoconductivity and direct bone bonding capacity, can produce early functional improvements in maxillofacial osteosynthesis of mandibular condylar fractures. However, to clarify the relationship between loading and mandibular condylar fracture healing with this plate system, it is necessary to evaluate the tissue over time. Further research reports are expected in the future.

The approach to mandibular condylar fracture is generally an extraoral approach [4]. Therefore, when it is necessary to remove the plate, there is a risk of recurrent facial nerve damage [31]. This is greatly stressful for both the surgeon and the patient. The bioresorbable system applied clinically may be an important option for the treatment of mandibular condylar fractures in the future.

The search for improved methods of subcondylar fracture fixation devices has been the subject of several studies. Mechanical loading tests are used to evaluate the behavior of fixation devices and methods, allowing the study of different osteosynthesis constructs. However, the present study did not involve a real mandible, and it will be necessary to conduct further research on mechanical experiments, as well as studies from various clinical viewpoints.

## 5. Conclusions

The titanium-based fixation system was much stronger than that based on bioresorbable u-HA/PLLA. In the biomechanical evaluation of mandibular condylar fracture treatment, however, our results showed that the u-HA/PLLA bioresorbable plate had sufficiently high resistance in the two-plate fixation method. The titanium plates were more resistant than the bioresorbable plates in subcondylar fractures when the initial force was applied in the anteroposterior direction; conversely, in the medial direction, there were no statistically significant differences. These results suggest that titanium and bioresorbable plate fixation systems have similar mechanical resistance within the limitation of our mandible model study.

## Figures and Tables

**Figure 1 materials-12-01557-f001:**
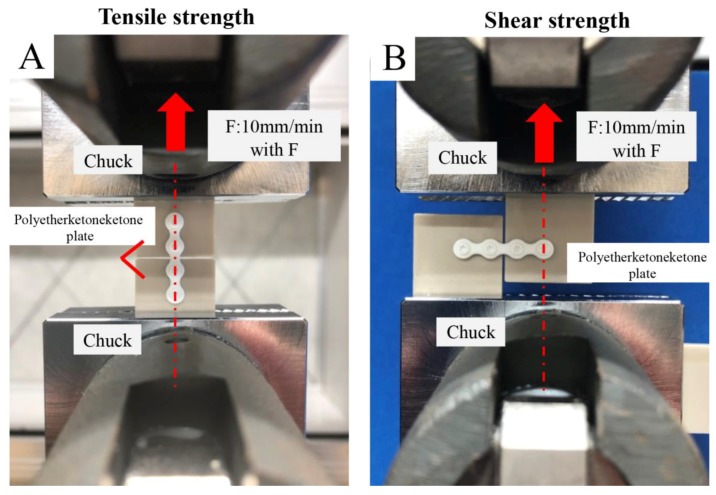
Mechanical strength models were prepared by fixing the plate with screws to the polyetherketoneketone plate. (**A**) Tensile strength; (**B**) shear strength.

**Figure 2 materials-12-01557-f002:**
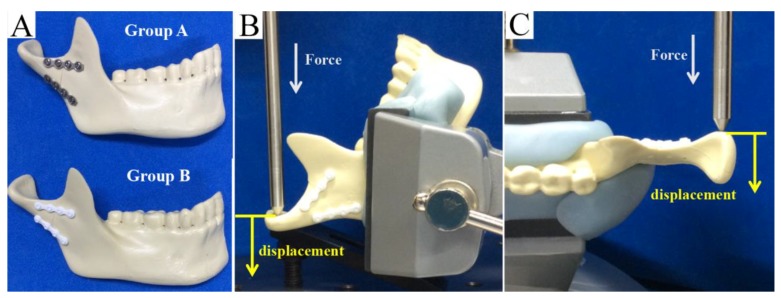
(**A**) The mandibular subcondylar fracture replicas were held in place by double titanium (Ti) straight plates (thickness: 1.0 mm) with four monocortical screws (2.0 mm in diameter and 6 mm long screws; group A) or by double unsintered hydroxyapatite (u-HA)/ poly-l-lactide (PLLA) straight plates (thickness: 1.4 mm), each with four monocortical screws (2.0 mm in diameter and 6 mm long; group B). A linear load was applied at a displacement speed of 1 mm/min. (**B**) Anteroposterior (vertical) linear loading. (**C**) Lateromedial (lateral) linear loading.

**Figure 3 materials-12-01557-f003:**
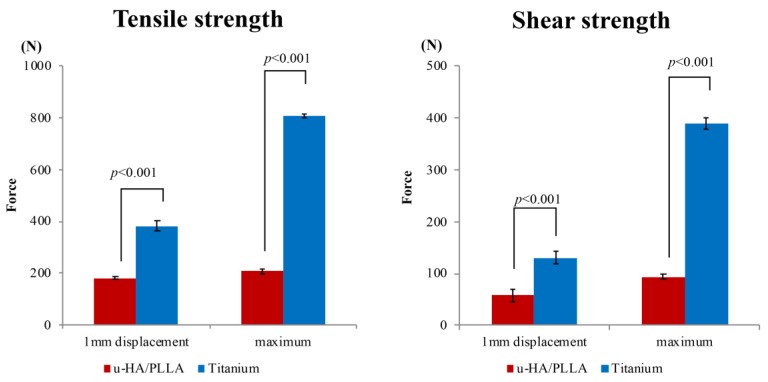
(**A**) Comparison of titanium plates and unsintered hydroxyapatite (u-HA)/poly-l-lactide (PLLA) bioresorbable plates with regard to maximum tensile test force and test force at 1 mm displacement in the tensile test. (**B**) Comparison of titanium plates and u-HA/PLLA bioresorbable plates with regard to maximum tensile test force and test force at 1 mm displacement in the shear test.

**Figure 4 materials-12-01557-f004:**
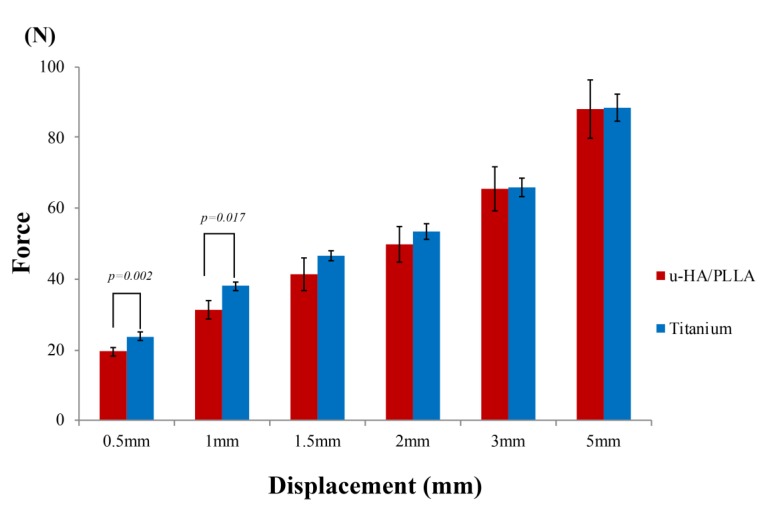
Load values of the titanium plates and unsintered hydroxyapatite (u-HA)/poly-l-lactide (PLLA) bioresorbable plates according to the amount of displacement in the anteroposterior loading test.

**Figure 5 materials-12-01557-f005:**
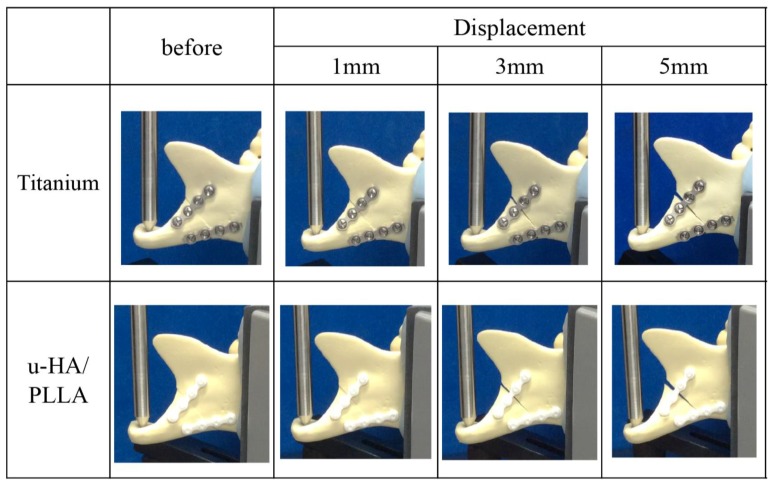
Change in subcondylar fracture segments under anteroposterior loading. Ti, titanium (plates); u-HA/PLLA, unsintered hydroxyapatite/poly-l-lactide (bioresorbable plates).

**Figure 6 materials-12-01557-f006:**
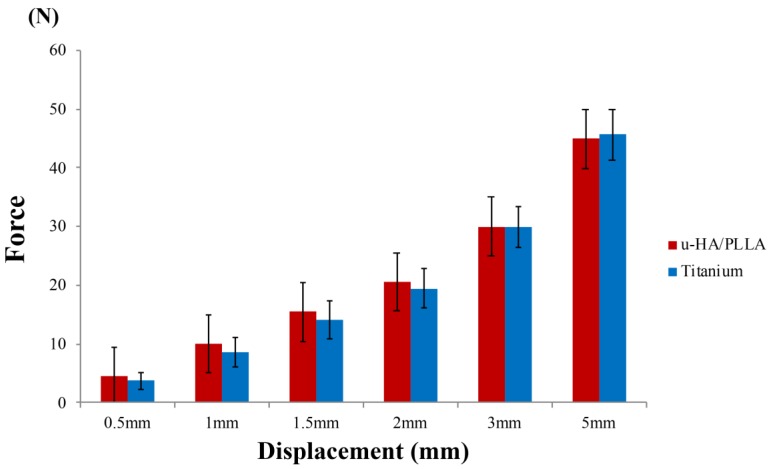
Load values of the titanium and unsintered hydroxyapatite (u-HA)/poly-l-lactide (PLLA) bioresorbable plates according to the amount of displacement in the lateromedial loading test.

**Figure 7 materials-12-01557-f007:**
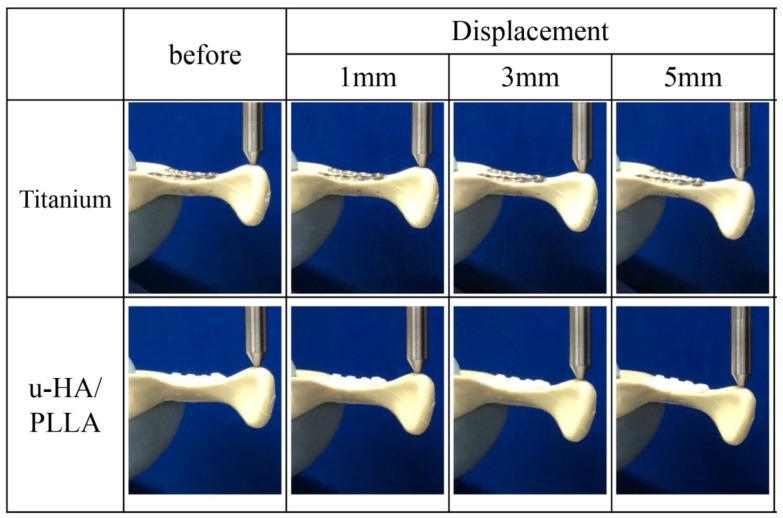
Change in subcondylar fracture segment under lateromedial loading. Ti, titanium (plates); u-HA/PLLA, unsintered hydroxyapatite/poly-l-lactide (bioresorbable plates).

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
