# Peer review of "Biomechanical Loading Comparison between Titanium and Unsintered Hydroxyapatite/Poly-L-Lactide Plate System for Fixation of Mandibular Subcondylar Fractures"

_materials, 2019, doi:10.3390/ma12091557_

Round 1
Reviewer 1 Report
In discussion, please, add a first paragraph with the summary of the study and the goal. Add also the limitation of your study.
Author Response
Reviewer 1:
In discussion, please, add a first paragraph with the summary of the study and the goal. Add also the limitation of your study. And we added the end of the discussion about the limitations of this study.
Author response: We thank the reviewer for these helpful comments. And, we added the summary and goals of this study to the first paragraph of discussion.

Reviewer 2 Report
The manuscript presents some interesting results concerning bio-resorbable and titanium based fixation systems for mandibular fractures.
In my opinion, the manuscript is interesting for a reader of Materials, but it needs major revisions.
Main comments
1) Strength measurements
The authors evaluated the tensile and shear strength of fixation systems based on bioresorbable composite plate in conjunction with screws, and titanium plate in conjunction with screws. These fixation systems were investigated using PEKK plates as substrates onto which plates were screwed (see Fig.1).
A) It is improper to report that the measured strength is that of the only plate, as the whole fixation system (plate and screws) was tested. Therefore, I suggest to change the following:
- Line 82: … by different osteosynthesis materials… Change with … by different osteosynthesis systems…
- Line 88: … osteosynthesis materials… Change with … osteosynthesis systems…
- Line 167: …in group A (titanium plates)… Change with …in group A (titanium plates fixed with titanium screws)
- Line 168: …in group B (u-HA/PLLA bioresorbable plates)… Change with …in group B (u-HA/PLLA bioresorbable plates fixed with bioresorbable screws)…
- Line 184: …The strength of the titanium plates was obviously higher than that of the u-HA/PLLA bioresorbable plates… Change with: …The strength of the titanium based fixation system was obviously higher than that of the u-HA/PLLA bioresorbable system…
- Line 227: …The strength of the u-HA/PLLA bioresorbable plates, in either tension or shear, was only about 45% that of the titanium plates… Change with: …The strength of the u-HA/PLLA bioresorbable system, in either tension or shear, was only about 45% that of the titanium system…
- Line 293: The titanium plates were much stronger than u-HA/PLLA bioresorbable plates. Change with: The titanium based fixation system were much stronger than that based on bioresorbable u-HA/PLLA.
B) Lines 92-93: The maximum stress and the stress at the time of 1-mm movement until the plate or screws were destroyed were measured.
This sentence is not clear and needs to be re-phrased. Also, the authors did not measure the maximum stress, they measured the maximum load.
Line 98 and line 101: The peak value of the profile… Change with: The peak value of the load profile…
C) The legend of Fig. 1: …the polycarbonate plate… Change with: the polyetherketoneketone plate…
D) Figure 1a and 1b: Into the figures it is reported F:10mm/min. This is improper. Change F:10mm/min with F or change F:10mm/min with v=10mm/min.
2) Biomechanical Loading Evaluation
This test represents the weakest point of this manuscript. The problem is the mandible model (see line 102) which is a polyurethane (PU) replicas of the mandible. Mechanical properties of PU do not reproduce properties of real mandibles. PU only match the properties of low density spongy bone. In other words, this model is too soft compared to real mandibles, it is a good model for training purposes, but not for biomechanical experiments. Results may be still of interest, but the authors need to critically interpret these results. For example, the gap shown between the fractured segments in figure 5 at 3mm or 5mm displacement (for both the composite and the titanium fixation systems) reflects the high compliance of the PU mandible model.
I suggest the following corrections:
- Lines 108-109: We used 20 polyurethane replicas of human hemimandibles with bonelike consistency, with a medullar portion and a cortical portion. Change with: We used 20 polyurethane replicas of human hemimandibles. Although polyurethane mandibles replicates the property of spongy bone [insert this reference: De Santis R, Sarracino F, Mollica F, Netti PA, Ambrosio L, Nicolais L. Continuous fibre reinforced polymers as connective tissue replacement. Composites science and technology. 2004 May 1;64(6):861-71.], this model is useful to obtain preliminary results concerning the stability of the investigated osteosynthesis systems.
- Line 109: (CHF 17.10 Mandible, Code #8900, SYNBONE AG, 109 Laudquart, Switzerland). Change with: (Mandible Code #8900, SYNBONE AG, 109 Laudquart, Switzerland). CHF 17.10 is the price of the model!
- Discussion section, lines 260-263. This opening… …plate group. Here the authors need to be critic and they need to specify that the polyurethane mandible model is softer than the real mandible, however PU replicates properties of spongy bone [use the previous suggested reference]. These results need to be considered as preliminary and further experiments are required to assess the difference between u-HA/PLLA and titanium fixation system.
- Conclusion section (lines 298-299). Also here the authors need to evidence that their results are based on a PU mandible model. Therefore, they have to specify that: within the limitation of our mandible model, results suggest that titanium and bioresorbable plates fixation systems have similar mechanical resistance.
Minor comments
Line 21: … the biomechanical intensity of… change with … the biomechanical strength of…
Line 23: To evaluate biomechanical loading... Change with To evaluate biomechanical behaviour...
Lines 26-27: This sentence is not clear.
Line 56: … absorbent plates, which are not required for plate removal,… Change with … resorbable plates, which do not require plate removal,…
Line 67: … the biomechanical intensity of… change with … the biomechanical strength of…
Line 76: A 2.0-mm miniplate system… Please, check the proper thickness. Line 90 reports 1.0 mm!
Lines 154-155: …displacement caused… this sentence is not clear. Please, rephrase.
Line 172: …differences were significant… Change with …differences were significant (p<0.05)
Line 186: …were significantly stronger… Change with …were significantly stronger (p<0.05)…
The labels of the y axis of figures 3,4,6 has to be changed: Newtons Change with Force (N)
Line 240: which is caused by traction of the lateral pterygoid muscle. A reference should support this statement, my suggestion is the following reference: Koolstra JH, Van Eijden TM. Dynamics of the human masticatory muscles during a jaw open-close movement. Journal of Biomechanics. 1997 Sep 1;30(9):883-9.
Author Response
Reviewer 2
The manuscript presents some interesting results concerning bio-resorbable and titanium based fixation systems for mandibular fractures.
In my opinion, the manuscript is interesting for a reader of Materials, but it needs major revisions.
Main comments
1) Strength measurements
The authors evaluated the tensile and shear strength of fixation systems based on bioresorbable composite plate in conjunction with screws, and titanium plate in conjunction with screws. These fixation systems were investigated using PEKK plates as substrates onto which plates were screwed (see Fig.1).
A) It is improper to report that the measured strength is that of the only plate, as the whole fixation system (plate and screws) was tested. Therefore, I suggest to change the following:
- Line 82: … by different osteosynthesis materials… Change with … by different osteosynthesis systems…
- Line 88: … osteosynthesis materials… Change with … osteosynthesis systems…
- Line 167: …in group A (titanium plates) … Change with …in group A (titanium plates fixed with titanium screws)
- Line 168: …in group B (u-HA/PLLA bioresorbable plates) … Change with …in group B (u-HA/PLLA bioresorbable plates fixed with bioresorbable screws) …
- Line 184: …The strength of the titanium plates was obviously higher than that of the u-HA/PLLA bioresorbable plates… Change with: …The strength of the titanium based fixation system was obviously higher than that of the u-HA/PLLA bioresorbable system…
- Line 227: …The strength of the u-HA/PLLA bioresorbable plates, in either tension or shear, was only about 45% that of the titanium plates… Change with: …The strength of the u-HA/PLLA bioresorbable system, in either tension or shear, was only about 45% that of the titanium system…
- Line 293: The titanium plates were much stronger than u-HA/PLLA bioresorbable plates. Change with: The titanium based fixation system were much stronger than that based on bioresorbable u-HA/PLLA.
Author response:We thank the reviewer for these helpful comments. As you pointed out, we corrected each part.
B) Lines 92-93: The maximum stress and the stress at the time of 1-mm movement until the plate or screws were destroyed were measured.
This sentence is not clear and needs to be re-phrased. Also, the authors did not measure the maximum stress, they measured the maximum load.
Author response:We thank the reviewer for this helpful comment. As you pointed out, we re-phased this sentence.
Line 98 and line 101: The peak value of the profile… Change with: The peak value of the load profile…
Author response:We thank the reviewer for this helpful comment. As you pointed out, we corrected it.
C) The legend of Fig. 1: …the polycarbonate plate… Change with: the polyetherketoneketone plate…
Author response:We thank the reviewer for this helpful comment. As you pointed out, we corrected it.
D) Figure 1a and 1b: Into the figures it is reported F:10mm/min. This is improper. Change F:10mm/min with F or change F:10mm/min with v=10mm/min.
Author response:We thank the reviewer for this helpful comment. As you pointed out, we corrected it and replaced the figure1.
2) Biomechanical Loading Evaluation
This test represents the weakest point of this manuscript. The problem is the mandible model (see line 102) which is a polyurethane (PU) replicas of the mandible. Mechanical properties of PU do not reproduce properties of real mandibles. PU only match the properties of low density spongy bone. In other words, this model is too soft compared to real mandibles, it is a good model for training purposes, but not for biomechanical experiments. Results may be still of interest, but the authors need to critically interpret these results. For example, the gap shown between the fractured segments in figure 5 at 3mm or 5mm displacement (for both the composite and the titanium fixation systems) reflects the high compliance of the PU mandible model.
I suggest the following corrections:
- Lines 108-109: We used 20 polyurethane replicas of human hemimandibles with bonelike consistency, with a medullar portion and a cortical portion. Change with: We used 20 polyurethane replicas of human hemimandibles. Although polyurethane mandibles replicates the property of spongy bone [insert this reference: De Santis R, Sarracino F, Mollica F, Netti PA, Ambrosio L, Nicolais L. Continuous fiber reinforced polymers as connective tissue replacement. Composites science and technology. 2004 May 1;64(6):861-71.], this model is useful to obtain preliminary results concerning the stability of the investigated osteosynthesis systems.
Author response:We thank the reviewer for this helpful comment. As you pointed out, we corrected it amd added the reference.
- Line 109: (CHF 17.10 Mandible, Code #8900, SYNBONE AG, 109 Laudquart, Switzerland). Change with: (Mandible Code #8900, SYNBONE AG, 109 Laudquart, Switzerland). CHF 17.10 is the price of the model!
Author response:We thank the reviewer for this helpful comment. As you pointed out, we corrected it.
- Discussion section, lines 260-263. This opening… …plate group. Here the authors need to be critic and they need to specify that the polyurethane mandible model is softer than the real mandible, however PU replicates properties of spongy bone [use the previous suggested reference]. These results need to be considered as preliminary and further experiments are required to assess the difference between u-HA/PLLA and titanium fixation system.
Author response:We thank the reviewer for this helpful comment. We added it as follows, 'However, the mechanical model used for this study was a polyurethane model. The polyurethane mandible model is softer than the actual mandible, so it should be noted that this results in the use of the resorbable plate system for clinical treatment immediately. '
- Conclusion section (lines 298-299). Also here the authors need to evidence that their results are based on a PU mandible model. Therefore, they have to specify that: within the limitation of our mandible model, results suggest that titanium and bioresorbable plates fixation systems have similar mechanical resistance.
Author response:We thank the reviewer for this helpful comment. As you point out, we added that it is a conclusion derived from the mandible model we used.
Minor comments
Line 21: … the biomechanical intensity of… change with … the biomechanical strength of…
Author response:We thank the reviewer for this helpful comment. As you pointed out, we corrected it.
Line 23: To evaluate biomechanical loading... Change with To evaluate biomechanical behaviour...
Author response:We thank the reviewer for this helpful comment. As you pointed out, we corrected it.
Lines 26-27: This sentence is not clear.
Author response:We thank the reviewer for this helpful comment. As you pointed out, we corrected this sentence. We changed to "A linear load was applied anteroposteriorly and lateromedially to each group to simulate the muscular forces in mandibular condyle fracture. "
Line 56: … absorbent plates, which are not required for plate removal,… Change with … resorbable plates, which do not require plate removal,…
Author response:We thank the reviewer for this helpful comment. As you pointed out, we corrected it.
Line 67: … the biomechanical intensity of… change with … the biomechanical strength of…
Author response:We thank the reviewer for this helpful comment. As you pointed out, we corrected it.
Line 76: A 2.0-mm miniplate system… Please, check the proper thickness. Line 90 reports 1.0 mm!
Author response:We thank the reviewer for this helpful comment. 2.0 mm indicates the screw diameter. Titanium plate systems are classified by screw diameter. In this research, the plate thickness is 1 mm as you pointed out.
Lines 154-155: …displacement caused… this sentence is not clear. Please, rephrase.
Author response:We thank the reviewer for this helpful comment. As you pointed out, we corrected to ‘All replicas were analyzed for 0.5-, 1.0-, 1.5-, 2.0-, 3.0-, and 5.0-mm displacements by loading and for the amount of displacement by the maximum load.’.
Line 172: …differences were significant… Change with …differences were significant (p<0.05)
Author response:We thank the reviewer for this helpful comment. As you pointed out, we corrected it.
Line 186: …were significantly stronger… Change with …were significantly stronger (p<0.05)
Author response:We thank the reviewer for this helpful comment. As you pointed out, we corrected it.
The labels of the y axis of figures 3,4,6 has to be changed: Newtons Change with Force (N)
Author response:We thank the reviewer for this helpful comment. As you pointed out, we corrected figure 3,4, 6.
Line 240: which is caused by traction of the lateral pterygoid muscle. A reference should support this statement, my suggestion is the following reference: Koolstra JH, Van Eijden TM. Dynamics of the human masticatory muscles during a jaw open-close movement. Journal of Biomechanics. 1997 Sep 1;30(9):883-9.
Author response:We thank the reviewer for this helpful comment. As you pointed out, we added sentences as function of lateral pterygoid muscles in this study.

Reviewer 3 Report
Thanks for your submission.
The method of your article is well designed, and this study would be helpful for the reduction of condyle using bioabsorbable plate.
This article can be acceptable to be published in this journal.
Author Response
Reviewer3:
Thanks for your submission.
The method of your article is well designed, and this study would be helpful for the reduction of condyle using bioabsorbable plate.
This article can be acceptable to be published in this journal.
Author response: Thank you so much for your very pleasant evaluation comment for our research in terms of study of biomechanical evaluation in treatment of mandibular condylar fracture of bioresorbable plate.
